# Nutritional and Bioactive Characterization of *Sicana odorifera* Naudim Vell. Seeds By-Products and Its Potential Hepatoprotective Properties in Swiss Albino Mice

**DOI:** 10.3390/biology10121351

**Published:** 2021-12-19

**Authors:** Silvia Caballero, Laura Mereles, Alberto Burgos-Edwards, Nelson Alvarenga, Eva Coronel, Rocío Villalba, Olga Heinichen

**Affiliations:** Facultad de Ciencias Químicas, Universidad Nacional de Asunción, San Lorenzo P.O. Box 1055, Paraguay; scaballero@qui.una.py (S.C.); aburgos@qui.una.py (A.B.-E.); nelson@qui.una.py (N.A.); ecoronel@qui.una.py (E.C.); rvillalba@qui.una.py (R.V.)

**Keywords:** *Sicana odorifera*, seeds, proximate composition, minerals, antioxidant activity, hepatoprotective, fatty acids, biowaste, by-products

## Abstract

**Simple Summary:**

This research highlights the prospect of kurugua seed by-product as a nutraceutical and functional food ingredient. Nutritional and bioactive profiling revealed that kurugua is rich in excellent nutritional compounds that can be exploited for human food development or in animal feed formulations. The seed by-product has shown great promise as an effective hepatoprotective agent and could be targeted for drug development.

**Abstract:**

The “Kurugua” (*Sicana odorifera*) is a native fruit that demonstrates attractive nutritional, coloring, flavoring, and antioxidant properties. The main by-products from the processing and consumption of kurugua fruit are epicarp and seeds. In this work, the properties of the seeds of *S. odorifera* were evaluated. The nutritional composition of the fruit seeds was determined through AOAC official methods and UHPLC-ESI-MS/MS profiling. The antioxidant activities were determined using in vitro methods, and the acute toxicity and hepatoprotective properties were investigated in Swiss albino mice. Quercetin derivatives and cucurbitacins were the main phytochemicals in the seeds’ methanolic extract and demonstrated some biological activities. GC-MS analysis revealed the essential fatty acids linolenic and linoleic as the main compounds present in seeds oil. The methanolic extract significantly reduced the serum levels of glutamic-pyruvic transaminase (GPT) and glutamic-oxaloacetic transaminase (GOT) in mice with induced hepatotoxicity (GPT *p* < 0.05; GOT *p* < 0.001) at the minor concentration tested (100 mg/kg EMSo). The results suggest that the *S. odorifera* seeds as by-products show potential use as a source of phytochemicals and in the production of oils with application in food supplements and nutraceuticals. Their integral use could contribute to waste reduction from kurugua fruits processing within the food safety and environmental sustainability framework.

## 1. Introduction

Current global challenges such as food safety, climate change, poverty, and health have a direct impact on the realization of the right to adequate food. 

Each challenge is negatively affected by food loss and waste, and developing sustainable global consumption and production systems is necessary [1]. Food by-products management has been recognized by the circular economy as one of the principal keys to reducing environmental and economic problems [2]. The recovery of bioactive molecules from the bio-residues or industrial by-products of fruits and vegetables has potential uses in the industrial sector. The processed fruits’ waste can be re-used and can lead to high-added-value products such as functional food ingredients, food coloring, novel pharmaceuticals for alternative therapies, and disease prevention [3]. 

The interest of the food and cosmetic industries for products from medicinal plants has increased, and it is necessary to broaden the investigation of natural sources, including fruit waste such as seeds [4]. The literature on value addition to fruit-derived waste is diverse. Overall, the extraction of bioactive compounds from fruit processing waste and the application of green methods for the valorization of these sources opens new avenues for food, chemical, and pharmaceutical industries, which have high potential, especially where availability of waste from fruit processing is abundant [5]. This current trend has led researchers to explore several methods to recycle waste for manufacturing new products, emphasizing green chemistry and greener processes. Melon by-products have been used as new feedstock for proteins’ recovery, employing biological precipitation; the cucumisin was separated from these by-products with carrageenan, an environmentally friendly process for the industries, which avoids solvents [2]. A novel bio-refinery approach would seek to produce a wider range of valuable chemicals from fruit processing waste. For instance, the residue from most of extraction processes could further be a renewable source of biofuels, and polyphenols may be useful as food products and pharmaceuticals preservatives [5]. Quercetin and quercetin-3-glycoside are being isolated from the fruit seed waste of papaya seeds. Grape seeds are rich in polyphenols, resveratrol, quercetin, and other flavonoids, which are confirmed to impart cardiovascular protective effects [6]. Natural flavoring agents such as limonene, pectin extraction from fruit peels, and the growing demand for natural products in the food and beverage industry show the scope of technological progress in this sector [5].

The year 2021 has been declared as the International Year of Fruits and Vegetables, encouraging populations to increase their consumption, seeking for the reduction of waste within the framework of Sustainable Food and Food Security, and valuing the potential of by-products, such as the seeds, not used in the industrial processing of fruits [7]. It has been reported that the processing of fruits or vegetables belonging to genera of Cucurbitaceae such as Cucumis (melon), Cucurbita (pumpkin), and Citrullus (watermelon) produce large amounts of waste and by-products, among them being discarded seeds. These by-products are an inexpensive raw material and a reliable source of bioactive phytochemicals, including antioxidant-rich polyphenols, tannins, flavonoids, and other components nutritionally important as essential fatty acids, dietary fiber, and minerals [8]. Cucurbitaceae seeds such as pumpkin, watermelon, and melon contain many nutrients such as protein, fibers, and minerals, highlighting their potential as a dietary supplement and a source of nutraceuticals; these potential applications contribute to the valorization of processing fruits by-products [9]. Pumpkin seed oil has been promoted as a new functional food and it is already produced and marketed as a healthy, edible cooking oil in some countries [10].

*Sicana odorifera*, or “Kurugua” is from the Cucurbitaceae family, found natively in the Latin American region, where it is widely used in folk medicine for various ailments; however, it is a species that has been hardly studied [4]. In the case of *S. odorifera* fruits, although it is not a fruit for mass consumption, it is precisely the lack of a market for its bio-waste that has limited its integral use, including the parts that represent the greatest loss, such as the peels (pericarp) and seeds (endocarp). Its crop is a strategy to increase food security and a family farming source [11]. The fruit itself has been used as a repellent, clothing perfume, or hot infusion with therapeutic uses in alternative medicine, inserted in popular wisdom at the Latin American level [12]. However, knowledge about the validated bioactive properties of these inedible parts (peel and seeds) is still limited. The black kurugua fruit (*Sicana odorifera* Naudim Vell.) is considered an exotic fruit; the large fruit contains 6% of its weight as seeds, in numbers from 900 to 1100 seeds per fruit. The epicarp of the fruit is intense purple, 30 to 60 cm in longitudinal diameter, and 9 to 15 cm in transverse diameter, and the pulp or mesocarp is approximately 2 cm thick, with oval seeds arranged in a row in the membranous endocarp. This bio-residue could have a great economic impact for the industrial processing of the pulp in the value chain, being, together with the peel, its main waste [11]. Studies on the use of its peel as a potential source of natural colorants have been published [13]. In *S. odorifera* seeds, insect repellent activity has been reported with aromatic properties. Triterpenes and flavonoids, including karounidiol dibenzoate, Cucurbita-5,23-diene-3h,25-diol, taxifolin, and quercetin were isolated from them [14]. The sweet aroma of the fruit, as well as the intense color of the peel, has been characterized [15,16,17]. The aromatic spectrum of the pulp is characteristic and the compounds responsible for the flavor were described (94.8% free volatile; with 61.1% as aliphatic alcohols [16], whereas studies on the composition of the seeds and the bioactivity of its components are still scarce. The phytochemical profile of the pulp is promising as a source of antioxidant compounds [18]. Regarding the peels, flavonols and anthocyanins with antioxidant activity have been described [17]. There are several animal models to evaluate the hepatoprotective effect of natural products; one of the effects is the model of liver damage induced by acetaminophen (APAP), a drug widely used as an antipyretic and analgesic, which in high doses can produce necrosis and insufficiency acute hepatica [19,20,21]. 

Liver disease is one of the leading causes of death in the world, and there is still an urgent demand for effective and safe hepatoprotective agents, despite advances in modern pharmacology [22]. The hepatoxicity of APAP is primarily caused by metabolism by cytochrome P450 to produce N-acetyl-p-benzoquinone imine (NAPQI), which can react with glutathione (GSH) to cause oxidative stress that can trigger the mitochondrial signal pathway and cause cell damage [23]. On the other hand, the solid evidence that demonstrates the multiple healthy effects of-3 PUFA for humans has stimulated the consumption of ω-3 PUFA. Its supply is limited, and is focused mainly on the consumption of fatty fish or bluefish and nutritional supplements based on fish oils or microalgae, thus hindering the increase in the consumption of these fatty acids in the western population. The incipient industrial production of vegetable oils rich in ALA in some Latin American countries is a novel and innovative alternative to increase the consumption and production of ω-3 fatty acids, specifically from its metabolic precursor, ALA [24,25]. Currently, plant residues in the form of *S. odorifera* seeds represent an opportunity to explore their properties and to give an integral use to industrialized fruits, following the current trend of generating more sustainable alternative processes, and the interest in the sustainable production of bioactive molecules [3]. The aim of this work was to describe the proximate composition, minerals and antioxidant activity of *S. odorifera* seeds and their fatty acids profile as well as the profile of polyphenol compounds, acute toxicity, behavior and hepatoprotective effect in mice of the methanolic extract, to explore their nutritional and nutraceutical potential in order to promote the use of this bio-residue.

## 2. Materials and Methods

### 2.1. Reagent and Standars

Acetaminophen and silymarin from Sigma Chemical Company (St. Louis, MO, USA) and sodium pentobarbital Nembutal (50 mg/mL) from Abbott (Osaka, Japan) were used; propylene glycol and ethanol were purchased locally. Kits for the estimation of alkaline phosphatase (ALP), glutamic pyruvic transaminase (GPT) and glutamic-oxaloacetic transaminase (GOT) were purchased from Human Diagnostics Worldwide reagent (Wiesbaden, Germany). Multi-element standard solution of Ag, Al, B, Ba, Bi, Ca, Cd, Co, Cr, Cu, Fe, Ga, In, K, Li, Mg, Mn, Na, Ni, Pb, Sr, Tl, Zn, anthrone reagent, D(+)- glucose monohydrate and BIOQUANT^®^ kit for total dietary fiber measurement were purchased from Merck (Darmstadt, Germany). The boron trifluoride–dimethanol complex 50–52% (Cat. B21357, L15847) was purchased from Alfa Aesar (Lancaster, England). Solvents n-hexane, methanol, acetonitrile and water HPLC and MS grade were purchased from J.T. Baker (Xalostoc, Mexico). HPLC-grade acetonitrile for liquid chromatography was purchased from Merck (Darmstadt, Germany). Petroleum ether residue analysis grade 40–60 and ABTS Biochemica reagent was purchased from AppliChem (Darmstadt, Germany). Standards of fatty acids mix 37 FAMEs, 10 mg/mL in dichloromethane were purchased from SIGMA-ALDRICH (Saint Louis, MO, USA) and Folin–Ciocalteu’s phenol reagent and gallic acid monohydrate ≤98% was from Sigma (Luxembourg, Germany). The acetylene 2.8 AA atomical absorption grade, analitical helium 5.0, nitrogen 4.6 FID and argon 5.0, were purchased from White Martins (Sao Paulo, Brazil). Other reagents were of analytical grade, purchased from Merck (Darmstadt, Germany).

### 2.2. Collection and Preparation of Samples 

The fruits of *S. odorifera* were harvested in mature state from a crop of *S. odorifera from* “Kurugua Poty” Foundation, San Lorenzo city, Paraguay (−25.3266340 N, −57.4832020 E). A herbarium material was prepared for botanical identification, registered as N° 57,234 at the Index Herbarium of Facultad de Ciencias Químicas, UNA. Once collected, fruits were refrigerated and transported to the laboratory where the seeds were immediately separated from the pulp. The fruits were cut manually into slices with a knife and the seeds were separated from the pulp. Later they were divided into two portions; one was lyophilized to determine the proximate composition, minerals, antioxidants analysis, hepatoprotective activity, and the UHPLC-DAD-MS profile. The other seeds were used for oil extraction and fatty acid profiling, with a manual press (CDr Food press, Florence, Italy). All experiments were carried out in triplicate. 

### 2.3. Seeds Extract Preparation 

For the acute toxicity and the hepatoprotective effect, methanolic extracts resuspended in distilled water were employed. The seeds were dried at room temperature (23–25 °C approx.) and reduced to powder (1700–425 µm) by grinding in a seed disintegrator (Severin, Sundern, Germany). Then 100 g of the powder was extracted with 750 mL of analytical grade methanol as previously described [26], with some modifications. 

Briefly, the powder was mixed with methanol in a bath (Lab Companion BS-06, Shanghai, China) at 30 °C for 90 min, stirring every 15 min. After standing overnight, the extract was filtered through a thin nitrocellulose membrane (125 mm). The residue was re-extracted twice under identical conditions, with the same volume of methanol. The filtered extracts were pooled and concentrated with a rotatory evaporator (Quimis Q344B1, Sao Paulo, Brazil) under reduced pressure at 40 °C. The extract was then stored in the dark and refrigerated at −4 °C until use. For UHPLC-DAD and UHPLC-ESI-MS/MS profiling, the dry residue was re-suspended with methanol at a 1:3 ratio (*w*/*v*).

### 2.4. Morphological and Physicochemical Characteristics of Seeds 

The yield of the seeds per fruit was determined with 10 fruits of the March 2021 harvest. The pulp (mesocarp), seeds, and peel (pericarp) were weighed separately using an analytical balance (Precisa XM-60 HR, Zurich, Switzerland). The water activity was determined in the pulp (Rotronic HygroPalm water activity system, Bassersdorf, Switzerland). The color of the seeds and pulp were measured with a colorimeter (ColorStay White Mårten GmbH 2020, Baden-Wuttermberg, Germany). The measurements were performed with a manual vernier caliper (Fowler, Auburndale, MA, USA).

### 2.5. Seeds Composition 

#### 2.5.1. Proximate Analysis and Mineral Composition 

For the proximate composition, AOAC official methods were used [27]. The determination of the caloric value was carried out by the AtWater method [28,29]. For the determination of soluble sugars in the seeds, the spectrophotometric method of Clegg’s Antron was used [30], with a glucose calibration curve (2–80 ng/mL). All analyses were performed in triplicate and recorded on a spreadsheet using Graph Pad Prism 8.3 software. 

#### 2.5.2. MeOH Extract of *Sicana odorifera* (Kurugua) through UHPLC-ESI-MS/MS Profile 

##### Chromatographic and MS Conditions of Seeds’ MeOH Extract Profile

Chromatographic analysis was performed on a UHPLC system (Waters, Milford, CT, USA), connected to an Acquity PDA eλ detector. The chromatographic procedure was carried out on a column of 2.1 mm × 100 mm, 1.7 µm (Phenomenex KINETEX core-shell EVO-C18, Torrance, CA, USA), maintaining a temperature of 40 °C. The mobile phase consisted of water (A) and methanol (B), both with 0.1% formic acid and 10 mM ammonium formate. The gradient elution program was as follows: 0–0.7 min, 20%–20% B; 0.7–3.2 min, 20%–40% B; 3.2–7.6 min, 40%–80% B; 7.6–8.3 min, 80%–100% B; 8.3–10.4 min, 100%–20% B, and 10.4–13.0 min with 20% B. The elution rate was 0.3 mL/min and the injection volume was 10 µL. The samples were dissolved with LC-MS degree MeOH at 10 mg/mL and filtered through 0.22 µm nylon syringe filters from a previous injection. The DAD chromatograms were recorded at 254 and 350 nm, while the UV spectra from 200 to 500 nm were acquired for peak characterization. The MS spectra were conducted on a Xevo TQD QqQ-MS mass spectrometer through an electrospray ionization (ESI) source. The ionization source was operated in negative and positive ionization modes. The MS conditions were as follows: source temperature; 150 °C; drying gas temperature; 500 °C; drying gas; N_2_ 999 K/h, flow rate, 900 L/h, cone gas flow; 50 L/h, cone voltage; 25 V; capillary voltage; 4000 V; collision energy; 30 V, collision gas; and Ar (3.5 × 10^−3^ mBar). The MS data were acquired employing MS scan mode within a range of *m*/*z* 80–800 (scan time 0.14 s), with a further selection of the major signals for MS/MS analysis with the daughter ion mode (scan time 0.14 s). The system was operated with the Waters Masslynx V4.1 software for data acquisition and qualitative analysis.

#### 2.5.3. Seeds’ Fatty Acids Profile by GC-MS 

To determine the fatty acid profile, the dried whole seeds were extracted with the manual hydraulic press (CDr Food press, Florence, Italy) at room temperature (23–25 °C), the pressed oil was centrifuged (Neofuge15R Heal Force, Shanghai, China) and it was derivatized to methyl esters following the International Standard 5509:2000 method, Animal and vegetable fats and oils—Preparation of methyl esters of fatty acids [31]. Briefly, 100 mg of oil was weighed into a 100 mL flat bottom flask and 4 mL methanolic sodium hydroxide (0.5 M) was added, refluxing the mixture at 60 °C until the oil droplets were completely disappeared (15 min approx.). Then, 3 mL of boron trifluoride (12%) in methanol was added and the reflux continued for total 33 min. Then, isooctane was added, and the mixture was removed from the hotplate after 1 min. The refrigerant was removed and 20 mL of a saturated NaCl solution was added and stirred vigorously for 30 s, 40 kHz at 40 °C, followed by another addition of this solution until the exact half of the balloon to separation of the phases. The isooctane phase was dried with a small amount of anhydrous Na_2_SO_4_. An aliquot of 150 µL of the isooctane phase was diluted with 750 µL of pure isooctane in a 1.5 mL amber vial for injection. A GC-MS QP2010 Plus from Shimadzu (Kyoto, Japan), equipped with an SPTM2560 Fused Silica Capillary Column 100 m × 0.25 mm × 0.32 µm film thickness from Supelco (Bellefonte, PA, USA), with ionization at 70 eV, was utilized, using He as the carrier. A volume of 1 µL of this solution was injected. The carrier gas was He (1 mL/min, 30–35 m/s). GC-MS conditions: The ionization mode was electron impact at 70 eV and the emission current was 100 μA. Temperatures were inlet at 220 °C, interface was 250 °C, injector temperature was 245 °C, and ion source temperature was 250 °C. The program started from 245 °C for 1 min, with a ramp of 80 °C/min to 300 °C. Split injection mode (split ratio 1:100). Standard curves for fatty acids were built for quantitation with different FAME concentrations within the 8–160 μg/mL range. For the identification of the peaks of the fatty acid methyl esters, an NIST17 library (Gaithersburg, MD, USA) was employed. Results were expressed as mg per 100 g of seeds.

### 2.6. Biological Assays

#### 2.6.1. Antioxidant Activity 

##### Determination of Total Phenol Content and ABTS Radical Inhibition Test

First, an ultrasound-assisted extraction (J.P. Selecta Ultrasons H-D, Barcelona, Spain) was carried out with methanol: water (60:40) and acetone: water (70:30). Then, the antioxidant activity was determined by the ABTS•+ cationic radical discoloration test [32,33]. The solution was prepared 24 h before the assay, and then it was diluted with absolute ethanol to reach an absorbance of 0.7 ± 0.02 at 730 nm (UV-1800, Shimadzu, Kyoto, Japan). A calibration curve of 6-hydroxy-2,5,7,8-tetramethylchroman-2-carboxylic acid (Trolox) was used. The wavelength used for the detection of ABTS radicals was 734 nm. The results were expressed as µM Trolox equivalents (TEAC)/g sample. 

##### Total Phenolic Compounds (TPC)

The same extract used for the previous section was used. Total phenolic compounds (TPC) were measured spectrophotometrically with the Folin–Ciocalteu reagent following the method described by Singleton and Rossi [34], where the blue-colored complex was quantified at 765 nm (UV-1800, Shimadzu, Kyoto, Japan). A gallic acid calibration curve was used. The results were expressed in mg of gallic acid equivalents (GAE) per 100 g of sample (mg of GAE/100 g).

##### Content of Total Vitamin C

The quantification of vitamin C in the samples was performed following the spectrofluorometric method 967.22 from the AOAC [27]. For the measurements, an L-ascorbic acid calibration curve (2.5–20 μg/mL) was built. The results were expressed in mg of vitamin C per 100 g of fresh weight of the pulp.

##### Determination of Monomeric Anthocyanins 

The determination of anthocyanins was carried out by the spectrophotometric method of differential pH, based on the color loss of the monomeric anthocyanins at pH 4.5 and the presence of color at pH 1, measuring at 510 and 700 nm [35]. The final concentration of anthocyanins (mg/100 g) was calculated based on the volume of extract and sample fresh weight. It is expressed in cyanidin-3-glucoside (MW: 449.2 and ε: 26,900).

#### 2.6.2. Effects of *Sicana odorifera* Seeds’ Methanolic Extract on Acute Toxicity and the General Behavior Test on Mice

Experimental animals: Swiss albino mice of both sexes, 30 ± 5 g body weight, from the Bioterium of the Facultad de Ciencias Químicas (UNA), were used to determine the LD50 (oral route), the activity on the general behavior, and the effect of *Sicana odorifera* seed extract on liver profile. In a 12/12-h light-dark cycle, the temperature of 22–25 °C and humidity of 50–60% were maintained inside the *Bioterium* and the experimentation room. The animals received commercial balanced food and drinking water ad libitum. They were fasted by the withdrawal of balance the night before each experiment. The extract samples were suspended in distilled water, preparing working solutions of 50, 100 and 200 mg/mL. Swiss albino mice were dosed in a stepwise procedure, by body weight (bw), using doses of 500, 1000 and 2000 mg/kg of body weight, p.o. of EMSo in different groups in search of LD50 [36].

The profile of the general behavior of albino mice under the influence of the extract of the seeds of *S. odorifera* was studied by direct observation. Groups of 5 mice of both sexes received orally (a) the blank (distilled water 0.1 mL/10 g of body weight) and (b) doses of 10, 100, 300, and 500 mg/kg of the extract of the seeds of *S. odorifera*, and the ethological parameters were recorded in an individual session and according to the group. Each animal was observed for 5 min with 30-min intervals within the 4 h. All mice were kept under brief daily observation for 7 days after administration of the extract. Both central and peripheral behaviors were recorded and conveniently tabulated according to their intensity where 0; no effect, 1; slight increase, 2; moderate increase, 3; intense increase and 4; very strong increase [37].

#### 2.6.3. Hepatoprotective Effect of the Methanolic Extract of *S. odorifera* Seeds

Swiss albino male mice were randomly divided into 8 groups (*n* = 6), and orally treated for 4 days: Blank group (distilled water), Vehicle group (V; 2.5% ethanol: 40% propylene glycol: 57.5% water); Acetaminophen group (APAP; water); Silymarin group (SM; 150 mg/kg of silymarin, p.o.); and Groups EMS10, EMSo100, EMSo300 and EM500 (treated with 10; 100; 200 and 500 mg/Kg of *S. odorifera* extract, respectively, p.o.). On the fourth day, acute hepatotoxicity was induced with acetaminophen (APAP), two hours after the oral treatment; all animals, except the ones in the blank group and the vehicle group, received 300 mg/kg, i.p., of APAP [38]. A total of three hours after APAP administration, the blood sample was collected by cardiac puncture after anesthesia with sodium pentobarbital (50 mg/kg, i.p.). The biochemical parameters of albumin, total proteins, alkaline phosphatase (ALP), glutamic pyruvic transaminase (GPT) and glutamic oxaloacetic transaminase (GOT) were determined from the serum. To obtain the blood serum, the fresh blood samples were incubated in a water bath for 20 min and then subjected to centrifugation for 15 min at 875 g. The samples were processed immediately after their collection. Control serum (normal and pathological Humatrol) was processed before each determination, as internal quality control, and the values obtained for the different biochemical parameters were always within the expected ranges. The measurements of albumin, total protein, ALAT, GPT and GOT were taken in a Biosystem BTS 350 semi-automatic analyzer (Barcelona, Spain).

### 2.7. Data Analysis

The data were recorded and processed in the GraphPad Prism 5.0 program (GraphPad Software Inc., San Diego, CA, USA). Descriptive statistics were applied. Results are expressed as means ± SD. For data analysis on biological assays, the statistical analysis used was one-way ANOVA followed by the Tukey test, and a confidence level of 95% was used for the comparisons between groups, where *p* < 0.05 was considered statistically significant. 

## 3. Results

### 3.1. Seeds Composition

#### 3.1.1. Proximate and Minerals Composition

The results of the physical properties, proximate composition, minerals, and caloric value of the pulp and seeds of *S. odorifera* fruits are shown in Table 1. The mature fruits show an oblong shape, dark purple color in the epicarp (peel), and orange color in the mesocarp (pulp). The mesocarp had a more intense lightness color than that of the seeds. The seeds were bicolor, both brown and beige. The main components of the seeds were lipids and dietary fiber, whereas the mesocarp showed more water and total carbohydrate. Thus, their caloric value was higher than that of the pulp. On the mineral composition of the seeds, potassium, was predominant, followed by magnesium and calcium, while zinc and calcium were major in the pulp (Table 1).

#### 3.1.2. Chemical Characterization through UHPLC-DAD and UHPLC-ESI-MS of the *S. odorifera* Seeds’ Extract

The MeOH extract of the Paraguayan kurugua (*S. odorifera*) seeds was analyzed employing UHPLC-DAD-ESI-MS/MS. The UHPLC-DAD profile at 254 and 350 nm are depicted in Figure 1, while the extracted ion chromatograms of each detected compound in negative and positive ion modes are shown in Figure 2 and Figure 3, respectively. The UHPLC-MS analysis allowed the tentative identification of 11 compounds in our samples, including one hydroxycinnamic acid derivative, one phenolic acid, four flavonols conjugates, three flavonol aglycones, and two cucurbitacins (Table 2). 

The tentative assignation of the detected compounds was based on their retention time (Rt), UV absorption maxima, pseudomolecular ions, and MS/MS fragmentation patterns, compared to literature when available (Table 2). 

The only hydroxycinnamic acid derivative was compound **1**, which showed a pseudomolecular ion at *m/z* 341, losing a hexose (162 amu) to afford a secondary ion at *m/z* 179, suggesting a caffeoyl moiety [39]. Therefore, compound **1** was tentatively assigned as caffeoyl hexoside. Compound **2** was tentatively identified as vanillic acid, based on its (M − H)^−^ ion at *m/z* 167 yielding and MS2 base peak at *m/z* 108; both characteristics of this phenolic acid. The UV absorption maxima around 260 and 290 nm supported the assignation [40]. 

Among flavonol conjugates, the peaks showed absorption maxima around 350 nm and an MS^2^ base peak at *m/z* 301; characteristics of a quercetin core [17]. Thus, compounds **3**, **4**, **5**, and **6** were tentatively identified as quercetin derivatives (Table 1). The first two displayed a pseudomolecular ion at *m/z* 609, losing a hexosyl rhamnoside moiety (308 amu) to yield the base peak at *m/z* 301 in both cases. However, the retention time was different, suggesting isomeric structures. Therefore, compounds **3** and **4** were tentatively assigned as quercetin hexoside rhamnoside 1 and 2, respectively. Peak **5** showed an (M − H)^−^ ion at *m/z* 579 and a neutral loss of a pentosyl rhamnoside moiety (278 amu) to afford the base peak at *m/z* 301. Thus, compound **5** was tentatively identified as quercetin pentoside rhamnoside [41]. 

The last (**6**) was tentatively assigned as quercetin rhamnoside because it exhibited a pseudomolecular ion at *m/z* 447 and a neutral loss of rhamnose (146 amu). The compounds **3** and **6** showed the most intense signals in the chromatographic profile (Figure 1), constituting the major compounds in the MeOH extract of *S. odorifera* seeds. 

Further, three other signals were detected in the kurugua seeds, in agreement with flavonol aglycones. The first peak (**7**) exhibited a pseudomolecular ion at *m/z* 301 and MS/MS fragments at *m/z* 151 and 107, compatible with quercetin [42]. Comparatively, the second peak (**8**) showed an (M − H)^−^ ion at *m/z* 285 and secondary ions at *m/z* 175 and 151, in agreement with luteolin. 

The last peak (**9**) displayed an (M − H)^−^ ion at *m/z* 329, losing a methyl (14 amu) to yield an intense fragment at *m/z* 314, suggesting the presence of quercetin dimethyl ether [43]. Therefore, compounds **7**, **8**, and **9** were tentatively assigned as quercetin, luteolin, and quercetin dimethyl ether, respectively. 

The analysis in the positive ion mode allowed the detection of two cucurbitacins. Their extracted ion chromatograms are shown in Figure 3. The first (**I**) showed an [M + H]^+^ ion at *m/z* 443 and was tentatively assigned as boeticol, while compound **II** with a pseudomolecular ion of 647 amu was compatible with karounidiol dibenzoate. The assignation of compound **I** was supported by the intense fragments at *m*/*z* 263 and 233, in line with cucurbitane-type triterpenoids [44]. Finally, the identity of compound **II** was supported by the MS/MS fragments at *m*/*z* 227, 171, and 105, characteristics of Karounidiol dibenzoate [45]. 

#### 3.1.3. Seeds Fatty Acids Profile by GC-MS 

The results of the analysis of the fatty acid profile of the seeds have shown that the fatty acids are preferably polyunsaturated (Table 3).

### 3.2. Biological Assays

#### 3.2.1. Antioxidant Activity 

The antioxidant activity was measured as a function of the TPC concentration, monomeric anthocyanins, and vitamin C, in addition to the total antioxidant capacity by inhibition of the radical ABTS, the results of which are shown in Table 4.

#### 3.2.2. Effects of *S. odorifera* Seeds’ Methanolic Extract on Acute Toxicity and the General Behavior Test on Mice 

Oral administration (500, 1000, and 2000 mg/kg) of the methanolic extract of *S. odorifera* seeds did not cause lethality in female mice after 24 h of observation.

After 14 days of observation, the mice were euthanized, and no signs of morpho-anatomical alteration were observed in the internal organs macroscopically evaluated and compared with the organs corresponding to the blank group. The methanolic extract of *S. odorifera* seeds administered orally up to a dose of 2000 mg/kg was shown to be safe because there was no lethality or symptoms indicative of acute toxicity under the experimental working conditions. 

No relevant effects on the general behavior were observed in mice of both sexes administered orally with different doses (10, 100, 300, and 500 mg/kg) of the extract. The most representative effects on the general behavior of the mice were observed within 4 h after administration and are recorded. The appearance of piloerection and self-cleaning behavior is observed in the group treated with the methanolic extract compared to the group treated with the blank. After this time, the influence on the parameters decreased until the total disappearance of these nonspecific effects, which were of a short duration and because they are the crude extract, are considered irrelevant.

#### 3.2.3. Hepatoprotective Activity Assay of *S. odorifera* Seeds’ Methanolic Extract

We have studied the activity of *S. odorifera* seeds’ extract on acetaminophen-induced hepatotoxicity. 

The influence of the oral administration of the seed extract on the serum levels of Albumin, total protein, and serum alkaline phosphatase of male mice with hepatic damage induced with acetaminophen is depicted in Figure 4. The oral administration of the methanolic extract of the seeds of *S. odorifera* did not show significant effects on the serum levels of alkaline phosphatase, total proteins, and albumin between the different treatments.

In the present study, it was demonstrated that the administration of acetaminophen, silymarin, and the different doses (10, 100, 300, and 500 mg/kg) of the methanolic extract of *S. odorifera* seeds (EMSo) did not produce a significant change in serum albumin, serum total proteins, and serum alkaline phosphatase values. On the other hand, Figure 5 show the influence of the oral administration of the extract on the serum levels of glutamic-pyruvic transaminase and Figure 6 show glutamic-oxaloacetic transaminase of male mice with hepatic damage induced with acetaminophen.

## 4. Discussion

According to the results, and other bio-residues, the seeds of *S. odorifera* have the potential for their use as a food ingredient in other prepared foods or for the extraction of oil from the seeds, whose lipid content is high (35.5%). Considering the RDI of manganese for healthy adults, which is 2.3 mg/day (Table 1), nutritionally, *S. odorifera* seeds can be considered an excellent source of manganese, because 100 g of the seeds can provide 1.8 mg, which is equivalent of up to 80% of the RDI for this mineral. Further, every 100 g of *S. odorifera* seed can provide up to 59% of the RDI for magnesium (RDI = 300 mg/day), 76.67% of the RDI for Cu (RDI = 0.9 mg/day), and more than 15% of the RDI of Fe (RDI = 14 mg/day); these results show that *S. odorifera* seeds can represent a good source of minerals. However, the mineral content and consequently the bioaccessible fraction may differ when it is released in the intestinal lumen after gastrointestinal digestion to be available for absorption, and further studies were suggested regarding the bioavailability of minerals from *S. odorifera* seeds [46,47,48]. 

The values of Vitamin C and TPC were lower than those reported in seeds of the same species, from fruits harvested in Colombia (16.0 ± 0.00 mg/100 g). However, the antioxidant activity values by the ABTS method in the seeds were higher than those reported (13.9 ± 2.87 µM TEAC/g) in the seeds of the ripe fruit, by the same authors [49]. In our previous work on pulp [13], we found lower concentrations of TPC in the mesocarp (37.2 mg GAE/100 g FW), whereas the epicarp (100 mg GAE/100 g FW) and dry seeds (206 mg GAE/100 g) showed higher levels of TPC [13,50]. The seeds, under our experimental conditions, showed 5.9 mg/g of cyanidin 3-glucoside in monomeric anthocyanins content. Compared to the other parts of the fruit, higher concentrations have been found in kurugua peel (19.7 mg/g of cyanidin 3-glucoside) and lower concentrations in the pulp (2.64 mg/g of cyanidin 3-glucoside) [13]. Anthocyanins are widely distributed metabolites also found in other Cucurbitaceae seeds [43]. Regarding the content of vitamin C in seeds, lower values than kurugua pulp (21.8 mg/100 g FW) were observed [50]. This was expected because vitamin C is water-soluble and, as we have seen, the seeds contain a high lipid content (35.5 g/100 FW). Finally, the total antioxidant activity was also measured by inhibition of the radical ABTS, where the detected compounds may participate in its scavenging. The total antioxidant capacity value found in the seeds (18.44 μM TEAC/g FW) of *S. odorifera* is higher than that reported for the pulp (4.39 μM TEAC/g FW) and skin (0.201 μM T/g FW) [13]. 

*S. odorifera* seeds presented a fatty acid profile, in which the main components were alpha linolenic acid (ALA) (omega 3) and linoleic acid (omega 6) (LA), followed by oleic monounsaturated fatty acid (omega9). ALA is an essential fatty acid for humans, which, when consumed in significant quantities, can be stored, β-oxidized, and metabolized into its bioactive derivatives, mainly DHA [51]. It should be noted that these analyses were carried out on samples from a harvest with average rainfall in the growing area of 1500 mm and that the concentrations of minor fatty acids may vary depending on environmental conditions. To verify the influence of these variables and the maturity of the fruit on the composition of the seeds, future studies are still necessary. Regarding the data on the fatty acid profile of the *S. odorifera* seeds, the presence of alfa linolenic acid (ALA) essential fatty acid omega 3 (C18:3) in 2.93 ± 0.01 mg per 100 g (Table 3) opens a field of work on the oil with food and industrial potential. In nutritional claims, if a food contains at least 0.6 g of ALA per serving (10 mL), it can be declared as a source of this compound. Lipids such as seed oil rich in linolenic and linoleic acids may be directly formulated as nutraceuticals for their proven health benefits [5]. The high content of this essential fatty acid in kurugua seeds can be exploited, using the wasted seeds as raw material for the extraction of ALA [52]. The main world supply of ALA is found in canola and soybean oils, whose contribution to the diet is quite limited compared to the linoleic acid content of these oils. Other sources of ALA are linseed oil (54%), chia (65%), sacha inchi (46%) and rosehip (26–37%) [53,54]. Similarly, kurugua seeds could be considered a food with a good source of ALA because they contribute to around 22–23% of the energy value of the 100 g of seeds, which is more than the 20% required [52,55]. In countries without a seacoast where bluefish consumption is low, and the supply of EPA and DHA omega 3 fatty acids are deficient, the diet-essential fatty acids come mainly from vegetable oils. To consider by-products as seeds as a source of these acids could be a good strategy. At the food industrial level, we know that there is a growing demand towards the development of foods with healthy oils that can replace critical ingredients such as trans fatty acids from partially hydrogenated vegetable fats, as has been established in world and regional regulation [25]. The goal of eliminating these oils is also in line with the global nutrition and diet targets related to chronic noncommunicable diseases (NCDs) established in the commitments of the United Nations Decade of Action on Nutrition (2016–2025), and with the draft Strategic Plan of the Pan American Health Organization 2020–2025 (OPS/OMS, 2019). Therefore, the oil from these seeds could be a good alternative for these purposes [24,56,57,58]. 

Regarding the UHPLC-MS analysis of the seeds, eleven compounds were detected under our experimental conditions. In a preview study of our group, twelve main compounds were tentatively identified, including five anthocyanin derivatives, five flavonols derivatives, two flavonol aglycones, and two unidentified compounds in *S. odorifera* peels [13]. In this work, two phenolic acids, compounds 1 and 2, as far as we know, are both described for the first time in *S. odorifera* seeds. On the other hand, vanillic acid was informed as one of the main components of *Cucurbita pepo* and *Citrullus colocynthus* seeds, other members of the Cucurbitaceae family [43,59,60]. In addition, caffeoyl glucose was identified in *Cucumis melo* seeds extract [43]. Regarding the detected flavonoids, mostly quercetin derivatives and luteolin aglycon were observed in our sample. The quercetin aglycon was isolated and fully characterized from the Colombian *S. odorifera* seeds [14], in agreement with our results. Further, quercetin hexoside rhamnoside was informed in several studies addressing its peels extract [13,17,18] and it is found in the seeds of *S. odorifera* for the first time. Similarly, luteolin, quercetin dimethyl ether, and quercetin rhamnoside had not previously been reported in *S. odorifera* seeds. However, luteolin was identified in other Cucurbitaceae members, such as pumpkin (*C. pepo*) seed oil [61] and bitter cucumber (*Citrullus colocynthus*) seeds’ extract [43]. Regarding quercetin dimethyl ether and quercetin rhamnoside, both compounds were detected in mouse melon (*Cucumis melo*) seeds’ extract, and the latter was also found in spiney gourd (*Momordica dioica*) seeds [43]. Finally, quercetin pentoside rhamnoside was previously observed in Chinese Plums (*Prunus domestica*) and has been described for the first time in *S. odorifera* seeds [41,62]. The cucurbitacins boeticol and karounidiol dibenzoate were previously isolated from the Colombian *S. odorifera* seeds, in agreement with our results [14]. In addition, karounidiol dibenzoate was previously informed in other Cucurbitaceae seeds [45,63], reporting interesting biological activities including anti-tumor and antiviral effects [45,64]. The plants from Cucurbita species are rich sources of phytochemicals and act as a rich source of antioxidants. 

In this work, the increase in transaminase levels induced by paracetamol is significantly reduced by the oral administration of EMSo (GOT < 0.001). Animals treated with 10 mg/kg EMSo presented significantly reduced levels of transaminases (GPT *p* < 0.05; GOT *p* <0.001); with 100 mg/kg EMSo (GPT *p* < 0.01; GOT *p* < 0.001); with 300 mg/kg EMSo (GPT *p* < 0.05; GOT *p* < 0.001) and with 500 mg/kg EMSo (GOT *p* < 0.001) compared to the APAP group (pathological control). In this work, in the mice treated with SM, used as a hepatoprotective control, the levels of GPT and GOT were significantly attenuated (*p* < 0.001) compared to the pathological group (APAP). It has been described that the protective effect of SM is associated with several possible mechanisms for antioxidant protection in the body, directly and indirectly; that is, these mechanisms directly involve the elimination of superoxide and prevention of radical oxygen species (ROS) formation on liver cells, but also indirectly the LM maintains an optimal redox balance by activating enzymatic and non-enzymatic antioxidants, and decreases inflammatory responses by inhibiting NF-kB pathways. Consequently, SM would reduce inflammation, necrosis, and transaminase levels. This leads us to think that a flavonoid such as SM that presents hepatoprotective effects could have mechanisms similar to those of the administered extract, considering that we have observed the presence of flavonoid compounds, mostly quercetin derivatives and luteolin aglycon [65,66]. It has been demonstrated that quercetin rhamnoside (Q7R), isolated from *Hypericum japonicum,* offers protection against CCl4-induced hepatotoxicity; this shows that the hepatoprotective effects of Q7R may be related to its antioxidant activity [67]. On the other hand, in the past several decades, the hepatoprotective activities of many natural triterpenoids have been reported from species of plants, such as cucurbitane triterpenoids from *Momordica charantia* (Cucurbitaceae), which showed cytoprotective effects against t-BHP-induced injury on hepatic cells, and cucurbitacin B isolated from the juice of *Ecballium elaterium* (Cucurbitaceae), which showed both preventive and curative effects against CCl4-induced hepatotoxicity by decreasing the abnormally increased GPT levels of mice [22]. In this work, the analysis in the positive ion mode allowed the detection of two cucurbitane-type triterpenoids; karounidiol dibenzoate and boeticol. Therefore, the presence of these cucurbitacins could also be involved in the hepatoprotective effect observed in the model used.

The activities of serum enzymes GPT and GOT are sensitive indicators of liver cell injury that may help recognize hepatic diseases. In this work, the results indicated (Figure 3) that administration of acetaminophen to the group that had received only distilled water significantly increased serum levels of biochemical parameters including GPT and GOT activities (group APAP) in comparison with the group that received the vehicle (*p* < 0.001). This demonstrates the hepatotoxic effect of 300 mg/kg APAP as a positive control for hepatotoxicity under the test conditions. In silymarin-treated mice (group SM), GOT and GPT were significantly attenuated (*p* < 0.001) when compared to the pathological group (APAP). It is difficult to predict which of the compounds observed in the methanolic extract profile contributed to the observed hepatoprotective effect. However, the content of polyphenols, specifically luteolin, quercetin, and four of their conjugated derivatives, may have some implications. In general, cucurbitacins, saponins, carotenoids, phytosterols, and polyphenols are the most important phytochemicals present in the cucurbits, and are bioactive phytoconstituents responsible for pharmacological effects such as antioxidant, antitumor, antidiabetic, hepatoprotective, antimicrobial, antiobesity, diuretic, and antigenotoxic activity [60]. This statement has been reinforced, pointing to the superiority of quercetin-O-glucoside over individual antioxidants in quenching free radicals during the on-line HPLC-DPPH• assay [68]. In this work, we have found the presence of quercetin and quercetin conjugated with hexoses in the seeds of *S. odorifera*. Quercetin, a bioactive secondary metabolite observed in methanolic extract, is incredibly important in terms of bioactivities as antioxidants, which has been proven by in vivo and in vitro studies. These properties of quercetin have been attributed primarily to their antioxidant capacity and free radical scavenging ability [6]. However, the hepatoprotective activity of Cucurbitaceae was reported, where a protein isolated from seeds from *Cucurbita pepo* showed hepatoprotective properties and caused a significant increase in the enzyme’s antioxidants, including CAT, SOD, and GPx, as well as glucose-6-phosphatase [60].

APAP is biotransformed to NAPQI, which conjugates with GSH and depletes it, then NAPQI binds to various enzymes, causing mitochondrial dysfunction and oxidative stress [23]. In this work, the presence of several flavonoids has been found, including quercetin, and luteolin, which prevent GSH depletion according to several studies [69,70,71]. 

APAP induces hepatocyte damage by different pathways, one of which is the increase in reactive oxygen species (ROS), which is the final result of a chain failure of the defense system from the oxidative stress of the hepatocytes. Some enzymes regulate this oxidative stress, such as glutamate-cysteine ligase (GCLC) and heme oxygenase-1 (HO-1), whose expression is regulated by the transcription factor Nrf-2, which is normally found in the cytosol, bound to its Keap-1 inhibitor protein. It has been shown that in the presence of APAP, the expression of Keap-1 increases, which is why it intervenes in this ROS regulation system [72]. One of the mechanisms of action of quercetin to prevent cytoxicity is to increase the translocation of Nrf-1 to the nucleus and thus increase the expression of GCLC and HO-1. This translocation is increased because quercetin binds to the same sites of action of Keap-1, therefore Nrf-1 is released. Furthermore, quercetin is capable of increasing the expression of the p-62 protein that competes with Nrf-1 for binding to Keap-1 [72]. APAP is also able to decrease the activity of superoxide dismutase (SOD), deplete the availability of GSH (antioxidant peptide), increase TNF-α, IL-6 (markers of inflammation), increase the amount of peroxynitrite (which initiates the necrosis), and induce endoplasmic reticulum stress. All these effects are counteracted by luteolin [69], a flavonoid that we have also found in the seeds of *S. odorifera.*

According to the current trend on the exploration of health-beneficial components from new resources, including by-products, the seeds of *S. odorifera* and their oil could be a valid alternative to be used in the production of natural new ingredients and bioactive compounds [10]. Despite the potential benefits, some points such as their safety, acceptability, and stability need to be considered for better exploitation of the resource. The presence and quantity of the bioactive compounds may vary with different crops seasons (due to crop management, environmental conditions, and maturity, etc.). Future studies on biological availability and mechanisms of action of the antioxidant compounds from kurugua seeds should be performed to fully understand their effect. Therefore, the development prospects of this seed oil and bioactive extracts also need to be further studied. Limited awareness about FPW in developing countries can be overcome with the help of academia and industry collaboration efforts [5]. Nevertheless, these results offer a baseline to advance following the SDG and the reduction of food losses, to pursue adequate processing, and for the use of waste from the food industry.

## 5. Conclusions

The nutritional characterization, chemical composition, and hepatoprotective activity of the kurugua seeds were studied. In general, they are oleaginous with 35% lipids and a high content of dietary fiber (34%). The oil is rich in polyunsaturated fatty acids, as the essential fatty acid linolenic (C18:3 ω3) is the major component.

The antioxidant capacity and phenolic content were comparable to other parts of the fruit. A total of eleven compounds were tentatively identified, including flavonoids; mostly, quercetin derivatives and luteolin aglycon were observed. Phenolic acids caffeoyl hexoside and vanillic acid, and flavonoids quercetin hexoside rhamnoside, luteolin, quercetin dimethyl ether, quercetin pentoside rhamnoside and quercetin rhamnoside, are described for the first time in *S. odorifera* seeds.

Furthermore, a hepatoprotective effect has been observed in the mouse model used in the methanolic extract of the seeds of *S. odorifera*. The seeds of this fruit contain bioactive molecules that are known for their beneficial effect on health and could contribute to the antioxidant and hepatoprotective activities observed. This work opens a path to follow in terms of possible applications of these seeds, a new source of healthy food with potential nutraceutical properties, and which currently constitute waste, in order to make full use of the fruit. 

## Figures and Tables

**Figure 1 biology-10-01351-f001:**
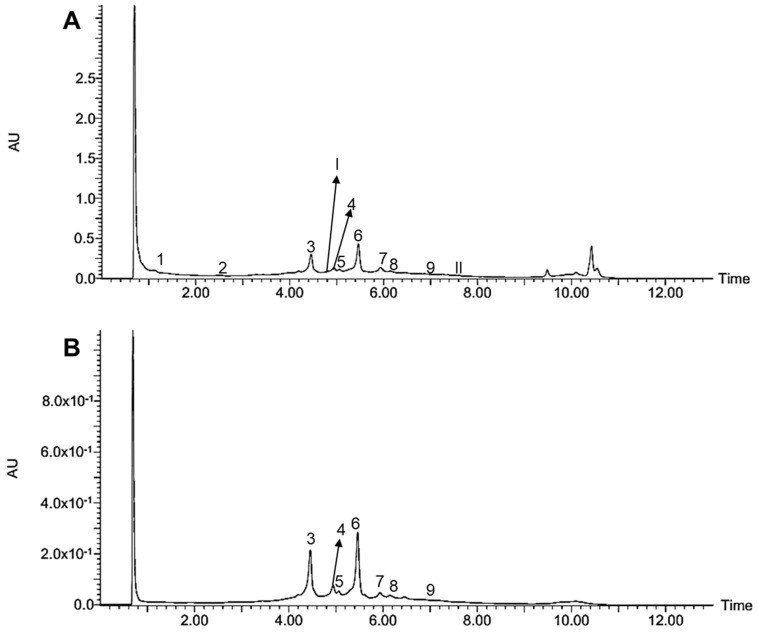
UHPLC-DAD profiles of the MeOH extract of *S. odorifera* seeds at 254 nm (**A**) and 350 nm (**B**).

**Figure 2 biology-10-01351-f002:**
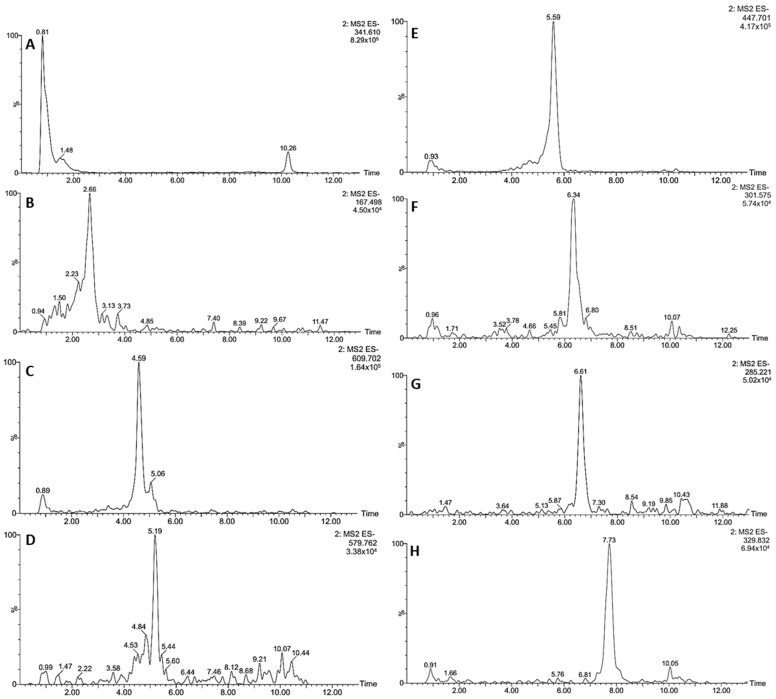
Extracted ion chromatograms in negative ion mode of the compounds tentatively identified in *S. odorifera* seeds’ MeOH extract: Caffeoyl hexoside (**A**), Vanillic acid (**B**), Quercetin hexoside rhamnoside 1 and 2 (**C**), Quercetin pentoside rhamnoside (**D**), Quercetin rhamnoside (**E**), Quercetin (**F**), Luteolin (**G**), Quercetin dimethyl ether (**H**).

**Figure 3 biology-10-01351-f003:**
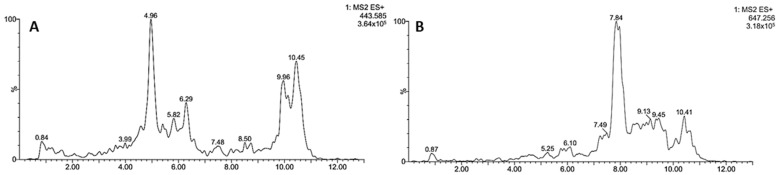
Extracted ion chromatograms in positive ion mode of the compounds tentatively identified in *S. odorifera* seeds’ MeOH extract: Boeticol (**A**), Karounidiol dibenzoate (**B**).

**Figure 4 biology-10-01351-f004:**
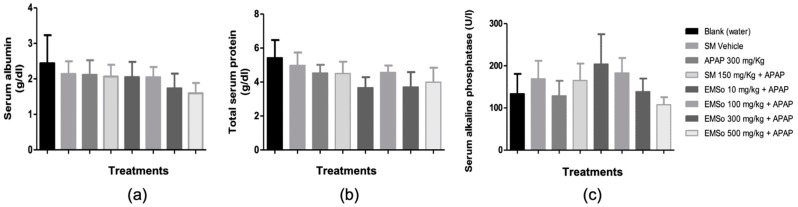
Influence of the oral administration of the seeds extract by group: (**a**) Variation of serum albumin levels of male mice in the different treatment groups; (**b**) Variation of serum levels of total proteins of male mice in the different treatment groups (**c**) Variation of serum alkaline phosphatase levels in male mice in the different treatment groups. Data are plotted as mean ± SD of six animals per group (*n* = 6). The statistical analysis used was one-way ANOVA followed by the Tukey test, where *p* < 0.05 was considered statistically significant.

**Figure 5 biology-10-01351-f005:**
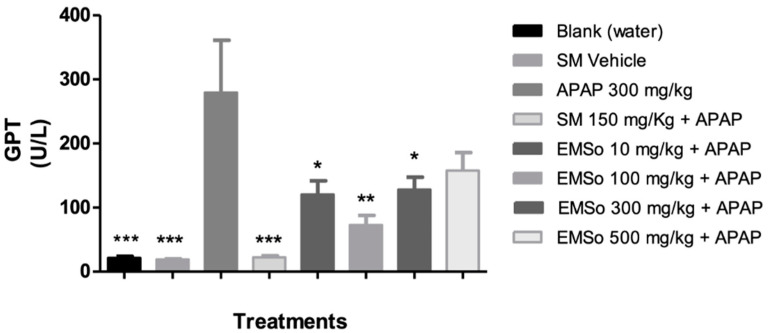
Influence of the oral administration of the seeds extract by group: Variation of serum levels of glutamic-pyruvic transaminase (GPT) of male mice in the different treatment groups. Data are plotted as mean ± SD of six animals per group (*n* = 6). The statistical analysis used was one-way ANOVA followed by the Tukey test, where *p* < 0.05 was considered statistically significant (* *p* < 0.05, ** *p* < 0.01, *** *p* < 0.001). SM; sylimarine, APAP; acetaminophen, EMSo; metanolic extract.

**Figure 6 biology-10-01351-f006:**
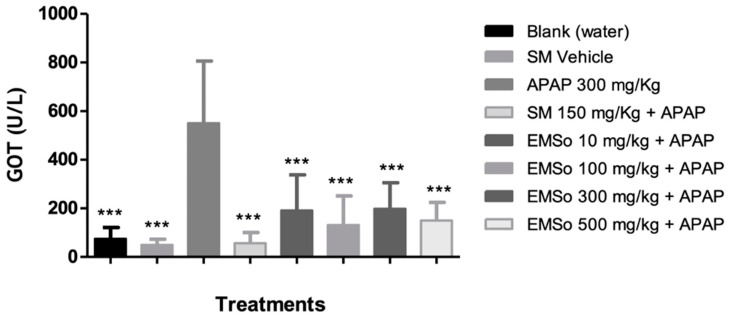
Influence of the oral administration of the seeds extract by group: variation of serum levels of glutamic-oxaloacetic transaminase (GOT) of male mice in the different treatment groups. Data are plotted as mean ± SD of six animals per group (*n* = 6). The statistical analysis used was one-way ANOVA followed by the Tukey test, where *p* < 0.05 was considered statistically significant (*** *p* < 0.001). SM; sylimarine, APAP; acetaminophen, EMSo; metanolic extract.

**Table 1 biology-10-01351-t001:** *Sicana odorifera* seeds and mesocarp physical characterization, proximate and minerals composition.

Parameter	Mesocarp (Fresh Weight)	Seeds (Dry Base)
Weight (g)	1970 ± 51	0.11 ± 0.01
Color	L* = 66.00 ± 2.45 a* = 11.71 ± 3.28 b* = 69.43 ± 2.32	L* = 34.80 ± 10.44 a* = 4.50 ± 5.52 b* = 8.70 ± 3.07
Longitudinal diameter (cm)	26.90 ± 1.4	1.58 ± 0.12
Transverse diameter (cm)	10.42 ± 0.7	0.80 ± 0.04
Water (g/100 g)	86.70 ± 0.4	10.06 ± 0.30
Total lipids (g/100 g)	1.31 ± 0.02	35.51 ± 0.40
Ash (g/100 g)	0.13 ± 0.01	2.55 ± 0.10
Total protein (g/100 g)	1.07 ± 0.08	18.05 ± 0.56
Total carbohydrate (g/100 g)	7.35 ± 0.31	2.80 ± 0.06
Dietary fiber (g/100 g)	3.11 ± 0.00	34.67 ± 0.31
Caloric value (Kcal/100 g)	45.5 ± 5	403 ± 5
Fe (mg/100 g)	0.35 ± 0.06	6.35 ± 0.69
Mn (mg/100 g)	0.55 ± 0.02	1.82 ± 0.10
Cu (mg/100 g)	0.25 ± 0.02	0.69 ± 0.09
Zn (mg/100 g)	42.31 ± 0.05	2.31 ± 0.09
Mg (mg/100 g)	5.16 ± 0.07	177.00 ± 4.66
Ca (mg/100 g)	29.61 ± 2.35	124.98 ± 9.17
Na (mg/100 g)	4.27 ± 0.54	29.51 ± 1.27
K (mg/100 g)	Nd	784.05 ± 52.40

The values are means ± DS (*n* = 10). Determinations made in fresh samples. Nd = no detected. * L: lightness, a* and b*; Coordinates that represent variation between reddish-greenish and yellowish-bluish, respectively.

**Table 2 biology-10-01351-t002:** Tentative identification of the compounds detected in the UHPLC-ESI-MS/MS profile of the MeOH seeds extract of *S. odorifera*.

Peak	Rt (Min)	UVmax	[M − H]^−^/[M + H]^+^	Polarity	MS/MS Fragments	Tentative Identification
**1**	1.22–1.48		341.61	Negative	179.48 (45), 119.14 (100)	Caffeoyl hexoside
**2**	2.66	290, 265	167.49	Negative	108.22 (100)	Vanillic acid
**3**	4.59	351, 265	609.91	Negative	608.75 (20), 300.68 (100), 179.27 (20)	Quercetin hexoside rhamnoside 1
**4**	5.06	350, 265	609.70	Negative	301.50 (100)	Quercetin hexoside rhamnoside 2
**5**	5.19	350	579.76	Negative	301.36 (100)	Quercetin pentoside rhamnoside
**6**	5.59	346, 265	447.96	Negative	300.46 (100), 271.19 (20), 255.11 (20), 179.29 (25)	Quercetin rhamnoside
**7**	6.34		301.58	Negative	150.92 (100), 107.37 (60)	Quercetin
**8**	6.61		285.22	Negative	175.08 (100), 151.22 (15)	Luteolin
**9**	7.73		329.83	Negative	314.78 (100)	Quercetin dimethyl ether
**I**	4.96		443.59	Positive	350.169 (30), 262.64 (65), 232.61 (100)	Boeticol
**II**	7.84–8.01		647.25	Positive	647.16 (85), 226.95 (87), 171.56 (40), 104.39 (100)	Karounidiol dibenzoate

* The compounds detected in positive ion mode are depicted in roman numerals (**I** and **II**).

**Table 3 biology-10-01351-t003:** *Sicana odorifera* seeds’ fatty acids GC-Ms profile.

Fatty Acids	Abbreviated Formula	mg/100 g
Miristic	C14:0	0.034 ± 0.02
Pentadecanoic	C15:0	0.01 ± 0.01
Palmitic	C16:0	3.64 ± 0.0
Palmitoleic	C16:1	0.02 ± 0.01
Margaric	C17:0	0.04 ± 0.01
Stearic	C18:0	2.33 ± 0.02
Oleic	C18:1c	4.32 ± 0.02
Linoleic	C18:2 ω6	9.98 ± 0.19
8,11 Octadecadienoic	C18:2 c	0.40 ± 0.01
Alfa Linolenic	C18:3 ω3	12.93 ± 0.01
Arachidic	C20:0	0.09 ± 0.01
Gondoic	C20:1	0.12 ± 0.01
	Total SFA	18.09 ± 0.09
	Total MUFA	13.10 ± 0.06
	Total PUFA	68.68 ± 0.574

The values are expressed as mean ± SD (*n* = 3). SFA: Saturated fatty acids, MUFA: Monounsaturated fatty acids, PUFA: Polyunsaturated fatty acids.

**Table 4 biology-10-01351-t004:** *S. odorifera* seed’s antioxidant potential.

Parameter	Seeds (SBS)
TPC (mg GAE/100 g FW)	47.34 ± 4.41
Monomeric anthocyanins (mg/100 g of cyanidin 3-glucoside)	5.90 ± 0.98
Vitamin C (mg/100 g)	1.11 ± 0.27
Total antioxidant capacity ABTS (μM TEAC/g)	7.47 ± 0.62

The values are means ± DS (*n* = 3). Determinations made in fresh samples. TPC: Total phenolic compounds.

## Data Availability

Not applicable.

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
