# Peer review of "Nutritional and Bioactive Characterization of Sicana odorifera Naudim Vell. Seeds By-Products and Its Potential Hepatoprotective Properties in Swiss Albino Mice"

_biology, 2021, doi:10.3390/biology10121351_

Round 1

Reviewer 1 Report

General comments:

This manuscript „ A source of potential nutraceutical compounds from a bio-residue, Sicana odorifera Naudim Vell. seeds”. Food processing by-products due to low costs and affordability are receiving tremendous interest among researchers all over the world as a potential strategy suitable for the production of bioactive ingredients and chemicals. Apparently, the title, abstract, and introduction have given the reader impression that authors have touched upon a very relevant and important issue addressed to valorizing of seeds recovered from a tropical fruit of Sicana odorifera Naudim Vell. to food and agricultural applications. However, after more detailed consideration its little academic value is revealed. The manuscript lacks a proper methodological section, while described methods cast doubt on the results obtained.

Specific comments:

The authors state that the seeds of Sicana odorifera due to the availability of nutrients and bioactives might represent commercial importance. Might be. However, the authors likely forgot that the object of their study is a natural product and therefore the accumulation of bioactives could vary from year to year depending on various factors, including maturity.  

Page 2, line 84. The authors wrote: “The fruits of S. odorifera were harvested in a mature state from the plantation that….”. I would like to ask the authors to specify what they mean by “mature state”, and how the above-mentioned state was measured. Whether there is any standardized maturity index as Streif, De Jager, or FARS indexes commonly used for apple harvesting? The maturity of fruit is very important since it has a great influence on the initial number of major compounds and bioactives.

Page 2, line 92. It is suggested to specify how the seeds were separated from the pulp and how the pulp was obtained?

Page 2, line 97. Please define room temperature.

Page 3, line 98. Provide the approximate size of powder particles.

Page 3, line 100. The pore size of filter used?

In my opinion, the authors should provide a full description of the reagents and standards used in this study. An additional section must be created.

Page 3, lines 145-146. Pressure?

Page 4, line 151. Whether 30 min was sufficient for complete transesterification of fatty acids present in oil fraction? What does the standard say?

Page 4, line 151. Please define the temperature of transesterification.

Page 4, line 162. Inlet temperature?

Page 4, line 170. Remove plus sign.

Page 4, line 170. Temperature, time, and kHz of UAE extraction must be provided.

Page 4, line 174. Please define the wavelength used for the detection of ABTS radicals.

Page 4, line 185. Total vitamin C.

Page 4, line 196. Please specify how the Sicana odorifera seed extract was served, i.e., dry, liquid. Amount of extract fed? Whether it has been incorporated into feed or water?

Sections 2.5.2. and 2.5.3. must be merged.

Page 5, line 215. Please provide more clarity on (0 to 4+).

Page 5, line 224. What “GPT”, “GOT” stand for? Please provide a description for each acronym the first time it appears in the manuscript. Go through the whole manuscript.

Page 5, line 229. 3000 rpm in g.

Page 6, line 243. Please check units for moisture.

Page 6, line 246. “Sicana odorifera” must be italicized.

Page 8, line 290. Poor separation of compounds, tailing, baseline drift. Likely behind peaks 3 and 6, there are other compounds. Such separation reveals bad chromatographic conditions.  In Figure 1 B there are no compounds 1 and 2. Please check.

Page 8, line 302. Oil polyunsaturated? I tend to think that rather fatty acids are unsaturated than oil. Check.

Page 9, line 303. “Table 3. Sicana odorifera seeds antioxidant potential.” Table 3 indicates the profile of FAME rather than AOA. Please check.

Page 9, line 310. “mgGAE/100g FW” I would suggest authors use the SI system throughout the whole manuscript. Please check.

Page 9, line 314. “16.0 ± 0.00 mg / 313 100g y”. Please check the unit.

Page 9, line 317 “total phenols” and page 9 line 318 “TPC”. The style used in this manuscript is confusing. I recommend authors be more consistent and use the same style within the whole manuscript.

Page 9, line 324. It is suggested to remove space between “values  than”

Page 11, lines 377-379. The authors wrote: “….S. odorifera have great potential for their use as a food ingredient in….” It is necessary to specify why the authors think that the selected material represents the potential for the food industry. Unfortunately, no antioxidant activity in a model system (TBARS, peroxide value) was demonstrated in the manuscript. Only one in vitro assay of the antioxidant activity using ABTS has been done in this study that rather is not enough to draw specific conclusions.  

Page 11, lines 379-382. interesting composition….. It is hard to say what exactly influenced the level of glutamic-pyruvic transaminase and glutamic-oxaloacetic transaminase since only crude extracts were tested in this experiment. Moreover, only qualitative analysis of the crude extracts has been done, while ignoring quantification of the compounds found.

Page 11, line 385. It is suggested to use the abbreviation for “Recommended Daily Intake” (RDI).

Page 11, line 395. Weird style of the reference “(CHAUDARI, 2017)” Please check.

Page 11, line 398. Hence, at least one more year of research needs to be done to check the influence of weather conditions and maturity stage on the content of fatty acids and other metabolites observed.

Page 14, line 527. I suspect that the authors misinterpret the results obtained. Currently, it is not possible to say unambiguously what compounds contributed to AOA and hepatoprotective activity the most. Probably the methanolic extracts contained some other compounds that also most likely promoted AOA and biological activity. I recommend omitting this sentence or rephrasing it.  

As the authors see, a lot of work needs to be done to make this article ready. Having addressed major concerns this work can be accepted.

Reviewer 2 Report

Thank you for giving me the opportunity to review this piece of work. In this study, Caballero et al. investigated the nutritional and bioactive composition of by-products from kurugua seeds. The authors subsequently examined its hepatoprotective properties in mice models. The study has the potential of sustainably promoting the valorisation of kurugua seed by-products as nutraceutical ingredient and potential drug discovery material that could promote health and well-being in humans.

Kindly, find below my comments for your perusal.

Title: The authors should please consider revising the title to convey the objective of the study. Consequently, I suggest they could consider “Nutritional and bioactive characterisation of Sicana odorifera Naudim Vell. seeds by-product and its potential hepatoprotective properties in Swiss albino mice”.

Summary:

The authors should consider rewriting the summary. I don’t get the use of “metabolic regulation” as that was not investigated. The authors should highlight the overarching goal of the research and its implication

Line 10: Kindly replace “are” with “is”.

Abstract

The authors should please revise the abstract. Line 13-15: This should be rearranged to “The “Kurugua” (S. odorifera) is a native fruit with attractive nutritional, coloring, flavoring, and antioxidants. The main bio-residues from the usage of these fruits are epicarp and seeds. In this work, the properties of the seeds of S. odorifera were evaluated.”

Line 15: what about the “composition?” Is it the “proximate and mineral” composition?

The determination of fatty acid profile using GC-MS should be stated as it remains a crucial part of the work.

Kindly, revise this “The composition of the fruit seeds was determined through AOAC official methods and UHPLC-ESI-MS/MS profiling, antioxidant and antimicrobial activities by in vitro methods, and the acute toxicity and hepatoprotective properties in Swiss albino mice.” The authors could present it as “The “nutritional” composition of the fruit seeds was determined through AOAC official methods and UHPLC-ESI-MS/MS profiling. The antioxidant and antimicrobial activities was determined using in vitro methods, and the acute toxicity and hepatoprotective properties investigated in Swiss albino mice”.

Line 18-20: Kindly revise this “The UPLC-MS profile of the seeds showed mainly quercetin derivatives and cucurbitacins, while a rich profile in essential fatty acids such as linolenic and linoleic was observed in GC-MS analysis”. I suggest you revise it to “Quercertin derivatives and cucurbitacins were the predominant bioactive compounds. GC-MS analysis revealed linolenic and linoleic as the main essential fatty acids present in the oils from the seed”. Also replace “profile” with “analysis”.

Line 20: By what magnitude of difference was the significant difference? Was it 50%? And what was the p-value?

Line 21: The authors use of the abbreviations (GOT) and (SGPT) are wrongly expanded.  The authors should correct it by matching the enzyme Glutamic oxaloacetic transaminase with the abbreviation (GOT) and serum glutamic pyruvic transaminase with (SGPT). They should also consider whether serum will be added to the (GOT) to make it (SGOT) just like they have indicated for the GPT enzyme.

Line 22-26: The authors should reduce the words highlighted for the implication of the work and rather bring out some of the other parts of the results such as the proximate and mineral contents.

Keywords: Currently, there are 11 words for the keywords. However, the journal allows up to 10 keywords. I want to suggest the authors delete “UPLC-DAD-Ms/Ms-Esi” as it is only an unexpanded abbreviation.

Introduction

Line 31-44: The authors have stated a long paragraph yet they support the paragraph with just one reference. There are several reviews that have been carried out on the valorisation of fruit seeds for high value products including nutraceutical agents or event for food/feed applications. The authors could introduce several of such works for the reader to know the advances made in this area of research. They could also highlight how the valorisation of such plant-based agro residues could contribute towards the attainment of the SDGs. These food by products create a nuisance on the environment as they contribute towards the release of greenhouse gases with the adverse effect of contributing towards climate change.

Line 45-54: Again, the authors continue to state a long paragraph that gets supported by a single reference. That is not appropriate. Meanwhile, a quick check online for the reference the authors have indicated brings out this paper “Technical and economic analysis for the production of banana and mango in Formosa” as reference [1]. This does not bear any close resemblance to the plethora of information on valorisation which the authors should highlight in the introduction.

Line 55: The authors report that the aroma of the fruit has been profile and the color has been investigated. The authors should kindly support that with references.

Line 68: The authors should delete one of the “by”.

My general impression about the Introduction is that the authors have to re-write it. The message is not systematically presented and makes the reader struggle to appreciate the background picture. Additionally, few references have been provided by the authors. The authors should please introduce citations to support the background information.

Materials and Methods

Line 87: What was the voucher number that was provided following the identification of the fruit?

Line 92-94: The authors should consider revising the statement. How was the pulp extracted? Was it through mechanical means? They could then say the seeds were harvested and divided into two; with one part lyophilized for the determination of centesimal and the other part subjected to a cold extraction with hexane to determine the fatty acid profile.

Line 97-103: What was the reference protocol the authors followed?

Line 98: The authors should expand the abbreviation “p.a.”. What was the amount of powders used for the extraction? What was the volume of methanol used for the extraction process?

Line 108: Which month in 2021 was the harvest done? Replace “in” with “using”. The fruits were weighed separately using………

Line 110: The authors forgot to provide “(city, country of origin)” as they have indicated for the Water activity meter.

Line 112: The authors should kindly indicate what measurements was carried out. Additionally, they should indicate “(city, country of origin)” for the Scali meter. Could the authors explain what the Scali meter is?

Line 114: The authors use “Centesimal”. A check online explains something different. The authors should consider replacing “Centesimal” with “Proximate analysis”.

Line 116: Kindly replace “At water” with “Atwater”. Atwater is one word.

Line 202: “chow” not “food”.

Line 126: Kindly delete “was”.

Line 131: Kindly revise “The samples were dissolved……”

Line 132: why “previous?”

Line 135: Kindly delete “and”

Line 145: replace extracted with “pressed” to obtain the oil. A full stop should be introduced.

Line 146: Start this sentence “The pressed oil was centrifuged (Italy) and it was derivatized to …..”.

Line 147: This “Animal and vegetable fats and oils - Preparation of methyl esters of fatty acids” should be removed.

Line 151: How much of the “Boron trifluoride” was added?

Line 152: How much of “isooctane” was added?

Line 207-210: What was the average weight of the mice? The authors have indicated mg/Kg of the extract that was administered. However, they failed to indicate how much of the extract per g or Kg of the body of the mice that was administered. Kindly indicate that.

Line 218: Is the water distilled water?

Line 228: What was the incubation temperature?

Data analysis

The authors failed to indicate how the data was analysed and which statistical software was used. This should precede the results section.

Results

Line 110-111: The authors report that “The color of the peel, seeds, and pulp were measured 110 with ColorStay colormeter White Mårten GmbH, 2020”. However, in the results section at Lines 237, the authors do not report the colour values in Table 1. I was expecting they will report the L*a*b* values as the colours that were measured and added to Table 1.  Also, why do the authors present most of the results in “decimals” but uses “commas” in few like the iron content of the mesocarp and the fibre content of the seeds. The authors should be consistent with the use of decimals. For example, to 2 decimal places as used for most of the values reported.

Line 243-244: The authors have titled Table 1 to include “Physicochemical characterization”. However, there is no report of physicochemical values. The “Weight and Diameters” reported are Physical values and not physicochemical values.

Line 246: The use of “UPLC-DAD and UPLC-ESI-MS profiles” is not correct. Those were the technical equipments that were used. The authors should rather use “Bioactive characterisation of……………”.

Line 254: What is “assignation?”

Table 3. The authors should kindly ensure consistency in the use of decimal places. Tables must stand alone so “ME” must be explained under the Table.

Line 312-330: The authors were discussion the results. However, there is a section on “Discussion”.

Figure 3. If the authors want use SGPT, same must be done for the GOT. Also, Figures must stand alone so abbreviations must be expanded. The authors have indicated that the statistical analysis carried out here. This should be under the section title “Data analysis”.

Line 334-335: The authors could support what they are putting across by showing pictures of the anatomical structures as revealed by a microscope.

Line 340: replace “in” with “on”

Line 351: The use of “Sicana odorifera” should be italiced in the title and must be applied throughout the manuscript.

Line 363: make “Acetaminophen” with “acetaminophen” and “Albumin” with “albumin”.

Figure 3…The authors must be consistent with keys used for the unit presentation of the extract quantity administered. mg/kg should be consistent.

Why is the APAP and the SM having a closed bracket for the concentration administered.

Line 373: replace “are” with “is”

Discussion

The beginning paragraph of the introduction must touch on the result presentation in Table 1. The physical properties, proximate and mineral composition and the rest…..”. The authors should discuss proximate composition as well.

Line 381: what is “hepatroprofector?”

Line 382: The authors should kindly check. The seed is rather rich in Magnesium and not Manganese. Kindly, check from Table 1.

Line 395: This “(CHAUDARI, 2017).” Must be deleted.

Line 396: replace “analyses” with “analyses”

Line 446-485: This was written without discussion and comparing it with what others have done. What could be causing the reduction in elevated enzymes levels reported? Could it be  some of the bioactive compounds that were identified? The discussion should be holistic. All analyses carried out must be brought to bear.

Line 479: The authors must add “…..and may help”

482: The authors must consistently use “decimals” in the p values and not commas.

Line 486: what is the “fact?”

Line 490-492: Why was this observation made?

Line 493-494: The authors should indicate the reference.

Conclusion:

The conclusion should address the objectives of the work. The authors failed to touch on the mineral composition yet it was investigated in this study.

Line 525: The authors should name some of the compounds.

Appendix:

Why have the authors put Figure 2 under appendix and Figure 1 in the text? I suggest they are all put in the text.

Round 2

Reviewer 1 Report

I see that the authors have addressed major concerns highlighted in my previous report. Now, the work can be accepted.

However, I would suggest to the authors make a correction to Figure 2 and Figure 3 providing labels for EIC plotted, i.e. Figure 2A,B,C,D, etc. Sufficient captions must be ensured.

Additionally, the authors must include references supporting the ability of quercetin to quench free radicals more effectively than other antioxidants. The authors may use the following sentence (lines 628-632): "....free radical scavenging ability [6]. This statement has been reinforced by  https://doi.org/10.1016/j.fbio.2020.100744, pointing to the superiority of quercetin-O-glucoside over other individual antioxidants in quenching free radicals during the on-line HPLC-DPPH• assay.”

Author Response

I see that the authors have addressed major concerns highlighted in my previous report. Now, the work can be accepted. 

However, I would suggest to the authors make a correction to Figure 2 and Figure 3 providing labels for EIC plotted, i.e. Figure 2A,B,C,D, etc. Sufficient captions must be ensured.

Thank you for your comments. As you suggested, each EIC from figures 2 and 3 was labeled and the figures’ captions were modified as well.

Additionally, the authors must include references supporting the ability of quercetin to quench free radicals more effectively than other antioxidants. The authors may use the following sentence (lines 628-632): "....free radical scavenging ability [6]. This statement has been reinforced by  https://doi.org/10.1016/j.fbio.2020.100744, pointing to the superiority of quercetin-O-glucoside over other individual antioxidants in quenching free radicals during the on-line HPLC-DPPH• assay.”

Thank you very much for your kind suggestion. We have edited the manuscript, please check.

Reviewer 2 Report

Thank you for responding to my comments. Kindly, find below some other comments on the revised manuscript for your response.

Reviewer comments:

Line 4: Kindly, replace “is” with “in”.

Simple summary: The authors should revise the summary. They can consider this “This research highlights the prospect of Kurugua seed by-product as a nutraceutical and functional food ingredient. Nutritional and bioactive profiling revealed that kuruagua is rich in excellent nutritional compounds that can be exploited for human food development or in animal feed formulations. The seed by-product has shown great promise as an effective hepatoprotective agent and could be targeted for drug development”.

Abstract:

Line 16: revise the sentence to this “…..that demonstrate attractive nutritional………..and antioxidant properties”.

Line 17: Revise to “The main by-products from the processing and consumption of Kurugua fruit are the epicarp and seeds.                                             

Line 20: replace “antioxidant activity” with “antioxidant activities”.

Line 22: revise the sentence to “……in the seeds and demonstrated some biological activities”.

Line 23: The sentence is not complete. “……….as the main what?”

Line 26-27: Revise this sentence “These results highlight the food potential from a bio-waste as non-traditional edible oil.”

Line 27: replace “represent” with “present”.

Introduction

Line 35: Introduce a “full stop” after adequate food.

Line 68-74: Break this sentence into two “It has 68 been reported that the processing of fruits or vegetables belonging to genera of Cucurbi-69 taceae such as Cucumis (melon), Cucurbita (pumpkin), and Citrullus (watermelon) pro-70 duce large amounts of waste and/or by-products among them the discarded seeds, are an 71 inexpensive raw material and a reliable source of bioactive phytochemicals, including an-72 tioxidant-rich polyphenols, tannins, flavonoids, and other components nutritionally im-73 portant as essential fatty acids, dietary fiber, and minerals [8].”

Line 75: Kindly, replace “pumpkin” for “Pumpkin”.

Line 127: replace “proximal” with “proximate”

Materials and Methods

Line 134: Correct it to “Reagent and standards”.

Line 159: Make it “A herbarium…….”

Line 164: kindly change the “proximal” to “proximate” and apply that throughout the manuscript.

Line 166: Replace “oils” with “oil”

Line 186: Kindly indicate the “model” of the analytical balance and the “city” it was produced.

The authors should kindly state the “Model” of the equipments used and the “City” they were produced and not rather the “City” and “Country”.

Line 194: For consistency sake, kindly change “centesimal” to “proximate”.

Data analysis

Line 325-326: This statement “All experiments were carried out in triplicate”, must be removed and placed under where the laboratory analysis were carried out.

Line 332: Kindly change the “Centesimal” to “Proximate” and let it apply throughout the manuscript.

Results

3.1.1. Centesimal and minerals composition:

Line 333-338: The authors must present the results in Table 1. All the parameters must be explained. What the authors have highlighted there is a bit different from what is in the Table. For example the authors could state that the mesocarp had intense lightness colour than that of the seeds. You can also add that potassium was the predominant mineral in the seed.

Table 3. The value of the “Pentadecanoic” is not in decimals. Kindly, make it in decimals for consistency sake.

Discussion

Line 491: replace “are” with “is”.

Line 525: Kindly replace “analyzes” with “analyses”.

Line 529: Revise this “However, this data on the majority fatty acid,……..”. You can consider this “However, regarding the data on the fatty acid profile of the S. odorifera seeds,……….).

Line 532-534: The authors should kindly re-write it.

Line 536: Replace “on” with “with”.

Line 541: Kindly, revise it

Line 586: Kindly, add p-value to this (GOT <0.001)

Line 503: Replace “are” with “were”. The use of tenses must be consistent.

Line 659: Revise this word “obtention”.
